# Green Synthesis of Narrow-Size Silver Nanoparticles Using *Ginkgo biloba* Leaves: Condition Optimization, Characterization, and Antibacterial and Cytotoxic Activities

**DOI:** 10.3390/ijms25031913

**Published:** 2024-02-05

**Authors:** Qi Ni, Ting Zhu, Wenjie Wang, Dongdong Guo, Yixiao Li, Tianyu Chen, Xiaojun Zhang

**Affiliations:** 1Key Laboratory of Resource Biology and Biotechnology Western China, Ministry of Education, Provincial Key Laboratory of Biotechnology, College of Life Sciences, Northwest University, Xi’an 710069, China; qini_0603@163.com (Q.N.); zt18228744069@163.com (T.Z.); 13298172926@163.com (W.W.); ddgyy2022@163.com (D.G.); cty20010314@163.com (T.C.); 2School of Medicine, Northwest University, 229 Taibai North Road, Xi’an 710069, China; yixiaoli010330@163.com

**Keywords:** green synthesis, *Ginkgo biloba*, silver nanomaterials, antimicrobial capacity, cytotoxic activity

## Abstract

Natural products derived from medicinal plants offer convenience and therapeutic potential and have inspired the development of antimicrobial agents. Thus, it is worth exploring the combination of nanotechnology and natural products. In this study, silver nanoparticles (AgNPs) were synthesized from the leaf extract of *Ginkgo biloba* (Gb), having abundant flavonoid compounds. The reaction conditions and the colloidal stability were assessed using ultraviolet–visible spectroscopy. X-ray diffraction, transmission electron microscopy, and Fourier transform infrared spectroscopy (FTIR) were used to characterize the AgNPs. AgNPs exhibited a spherical morphology, uniform dispersion, and diameter ranging from ~8 to 9 nm. The FTIR data indicated that phytoconstituents, such as polyphenols, flavonoids, and terpenoids, could potentially serve as reducing and capping agents. The antibacterial activity of the synthesized AgNPs was assessed using broth dilution and agar well diffusion assays. The results demonstrate antibacterial effects against both Gram-positive and Gram-negative strains at low AgNP concentrations. The cytotoxicity of AgNPs was examined in vitro using the CCK-8 method, which showed that low concentrations of AgNPs are noncytotoxic to normal cells and promote cell growth. In conclusion, an environmentally friendly approach for synthesizing AgNPs from Gb leaves yielded antibacterial AgNPs with minimal toxicity, holding promise for future applications in the field of biomedicine.

## 1. Introduction

Nanomaterials, known for their distinctive physicochemical properties, have diverse applications in drug resistance diagnosis, infection control, and the design and delivery of antimicrobial agents [1]. Numerous studies focused on exploring the antimicrobial activity of metal nanoparticles (NPs), including AgNPs, SeNPs, AuNPs, TiO_2_ NPs, single-walled carbon nanotubes, lysozyme-ZnO NPs, Au@Bi_2_S_3_ NPs, and Al_2_O_3_ NPs. These NPs can act independently or in conjunction with other molecules and achieve enhanced bacteriostatic properties. In contrast to traditional antibiotics, metal NPs have a wide range of target microorganisms, thereby reducing the likelihood of developing resistance [2]. Among these metal NPs, silver nanoparticles (AgNPs) are widely known for their potent broad-spectrum antimicrobial effects. Their efficacy has been extended to various bacteria, viruses, and fungi [3]. Compared with other metals, AgNPs have extensive biomedical applications, such as antibacterial and anticancer treatments, drug delivery, wound repair, vaccine adjuvants, medical dressings, and biosensing. Treatment of chronic infections is a significant challenge in antimicrobial therapy. AgNPs tend to form aggregates, thereby affecting their performance. Controlling the reduction and stabilization are two fundamental elements that contribute to obtaining stable colloidal AgNPs, which is advantageous for their biological activity [4].

Three broad strategies are currently used to synthesize nanosilver: chemical, physical, and biological. However, chemical and physical methods have limitations, such as high energy consumption, being non-environmentally friendly, low biocompatibility, and the need for toxic chemicals. Furthermore, chemical reduction methods often require the addition of stabilizers to prevent the agglomeration of AgNPs [5]. Biosynthesis has emerged as an eco-friendly alternative technology for synthesizing nanosilver. It utilizes natural biological resources, such as vitamins, sugars, plant extracts, biodegradable polymers, and microorganisms. Natural resources provide a rich and diverse range of easily accessible and cost-effective compounds [6]. Specifically, plant-mediated synthesis is advantageous because of its high reduction and stability, making it suitable for large-scale production [7]. Shivakumar et al. successfully synthesized AgNPs with effective antibacterial properties by utilizing *Eucalyptus* wood hydrolysate, obtained from the paper industry, at room temperature [8]. In contrast to chemical reduction methods, the plant-mediated synthesis of AgNPs does not require external stabilizers. Various plant compounds, including proteins, phenolic compounds, terpenoids, flavonoids, and polysaccharides, have been identified as reducing and capping agents that promote the stabilization and biocompatibility of AgNPs. Furthermore, phytochemicals that adhere to the surface of NPs synergistically enhance their antibacterial and anticancer activities [9]. Some successful syntheses of AgNPs have been reported using plants such as *Artemisia absinthium*, chamomile, pomegranate, acacia, longan, and green tea. Although different plant parts can synthesize metal NPs, most bioactive components with reducing properties are concentrated in the leaves, making plant extracts the preferred choice for synthesis [10]. However, different plants significantly affect the physical properties of NPs due to content variations in their reducing components [11]. The size of AgNPs is correlated with their antimicrobial activity; smaller particles allow direct contact with the bacterial cell wall. As silver ions cross the microbial membrane, their gradual release disrupts cell metabolism and eventually causes lysis. Among the ~374,000 plant species worldwide, 28,187 have potential applications in human medicine, making medicinal plants a preferred source for green synthesis and expanding their biomedical applications [12]. Moreover, the synergistic antibacterial effects of plants in combination with AgNPs have also been reported [13].

*Ginkgo biloba* (Gb) is an ancient Chinese tree that belongs to the Ginkgo family. Its use as traditional herbal medicine in China has a long history. The leaf extract of Gb (EGb) has various therapeutic effects, including antioxidant and anticancer effects, inflammation control, improvement of dementia symptoms, promotion of osteogenesis, and prevention and treatment of cardiovascular diseases [14,15,16,17]. Bioactive components mainly present in Gb, such as flavonoids and terpenoids, may play key roles in stabilizing AgNPs [18]. Currently, there have been studies that synthesize AgNP particles from Gb leaves [19]. However, the influence of synthesis parameters on the nanoparticles’ characteristics has been inadequately considered in the literature. In addition, not all studies focus on the antibacterial aspects [20]. Moreover, there is insufficient evaluation of the nanoparticles’ cytotoxic effects on various cell types. In the present study, in order to optimize the synthesis of AgNPs using EGb, we investigated the effects of various synthesis parameters, including reaction time, temperature, pH, and concentration. By controlling these key factors, we were able to achieve an efficient and reliable synthesis of AgNPs. AgNPs were characterized using analytical techniques such as ultraviolet-visible (UV) spectroscopy, transmission electron microscopy (TEM), X-ray diffraction (XRD), and Fourier transform infrared spectroscopy (FTIR). The antibacterial activity of AgNPs was tested against *S. aureus*, *E. coli*, *A. baumannii*, and *P. aeruginosa*. The toxicity of the green-synthesized AgNPs in different cell types was also evaluated. By combining the advantages of green synthesis with the potential bioactivity of Gb, our research provides insights into the sustainable production of AgNPs and their potential applications in antibacterial and biomedical fields.

## 2. Results and Discussion

### 2.1. Characterization of Metabolites in the Leaf Extract by HPLC-MS Analysis

The HPLC-MS results of secondary metabolites in Gb leaves are presented in Figure 1 and Table 1. Thirty plant compounds were identified and listed based on ion intensity, including flavonoids (e.g., Kaempferol-3-O-rutinoside isomer, Chrysin 7-gentiobioside, Isorhamnetin-3-O-rutinoside, Quercetin-3-O-rutinoside, and Typhaneoside), phenolic acids (e.g., Quinic acid), terpenoids (e.g., Blinin, Ginkgolide B, and Ginkgolide C), and carbohydrates (e.g., Sucrose). Flavonoids constitute the largest proportion of the identified components and potentially play a predominant role in AgNP synthesis. Flavonoids and terpenoids belong to polyphenolic compounds, which are secondary metabolites widely present in plants and possess redox properties based on phenolic hydroxyl groups. The hydroxyl groups of polyphenols can act as hydrogen bond donors and acceptors, facilitating interactions with bioactive components or nanomaterials. The reactivity of phenolic compounds is associated with the energy required for electron transfer. A linear relationship has been reported between polyphenol concentration and AgNP synthesis, with compounds containing adjacent OH groups exhibiting higher reactivity compared to monophenols. Plant extracts rich in flavonoids are advantageous for AgNP synthesis, as the hydroxyl and carbonyl groups in these compounds participate in the chelation of Ag ions and provide a stabilizing covering for AgNPs, resulting in the formation of smaller and more stable NPs. Terpenoids contain alcohols, ketones, aldehydes, and lactones that participate in the reduction and stabilization of AgNPs. Terpenoids can adsorb onto the surface of metal NPs through π-electron or carbonyl interactions or form organic matrices to temporarily prevent NP aggregation [21]. The hydroxyl and carboxyl functional groups present in carbohydrates can prevent the aggregation of nanoparticles through intramolecular hydrogen bonding [22]. Through hydrophobic interactions, hydrogen bonds, and covalent bonds, proteins can form protein–polyphenol complexes that protect proteins from hydrolysis and denaturation and stabilize NPs [23]. While Gb leaves only contain trace amounts of phenolic acids, flavonoids, and terpenoids may play a dominant role in AgNP synthesis, while proteins are essential for NP stabilization [24]. The complex chemical composition of plant ligands, which possess multiple metal chelation sites, facilitates the interaction between these ligands and the surface of Ag^+^ ions. As a result, the formation of cross-linked and extendable complex networks contributes to the stability of nanoparticles [25].

### 2.2. AgNP Synthesis

The typical synthesis of NPs can generally be divided into three stages: nucleation, evolution of nuclei into seeds, and growth of seeds into nanocrystals [26]. The generation of nanosilver was initiated by adding EGb. Figure 2 depicts a schematic diagram of the synthesis process of AgNPs. During stirring, the color of the solution changed from pale yellow to a bright reddish-brown, indicating that the reducing components in the plant extract reduced the Ag^+^ ions in AgNO_3_ to Ag atoms (Figure 2A). In addition, because of their high molar absorbance and bright color, AgNPs are suitable for colorimetric detection. UV spectrophotometry was used to investigate the optical properties that mainly depend on the size effects to identify the formation of metal NPs. The UV spectrophotometry results showed that the characteristic surface plasmon resonance (SPR) band appeared in the visible region at ~420 nm (Figure 2B), demonstrating the successful synthesis of AgNPs. AgNO_3_ and plant extracts do not exhibit the same peaks. AgNPs synthesized from various plant extracts have variable absorption peaks, but their SPR bands are typically located between 350 and 450 nm [27,28]. Therefore, our results are consistent with those in the literature. Furthermore, the appearance of individual SPR bands indicates the emergence of isotropic particles [29]. The specific conditions and outcomes of AgNP synthesis are presented in Table 2.

#### 2.2.1. Effect of Silver Nitrate Concentration

The effect of different AgNO_3_ concentrations on AgNP synthesis is shown in Figure 3A. The UV spectrum exhibits the highest peak at an AgNO_3_ concentration of 6 mM, indicating optimal synthesis conditions. This can be attributed to the nucleation of NPs. As the Ag salt concentration increases, a portion of the Ag^+^ ions are reduced to Ag atoms by the plant extracts. These Ag atoms serve as nucleation centers, facilitating the reduction of the remaining Ag^+^ ions and generating more AgNPs, thus increasing absorbance. The individual nuclei continue to grow, form clusters, and result in an increase in the NP size. This growth also causes a red shift in the UV spectroscopy absorption peaks towards higher wavelengths. The position of the SPR peak strongly depends on the concentration of Ag^+^. Within a concentration range of 4–8 mM AgNO_3_, an increase in Ag^+^ does not alter the position of the SPR peak. Previous studies have demonstrated that the aggregation effect and competition among NPs contribute to the shift in the SPR band, independent of the increase in Ag^+^ concentration [30]. In addition, treatment with 10 mM AgNO_3_ led to the aggregation of NPs due to the increased collision frequency of Ag^+^ ions. When exposed to high concentrations of AgNO_3_, insufficiently reacted silver salts are deposited on AgNPs, resulting in an unclear surface [31]. Therefore, among the AgNO_3_ concentrations tested in this study, 6 mM is the most suitable concentration for synthesizing AgNPs.

#### 2.2.2. Effect of the EGb Extract Concentration

The effect of the EGb extract concentration on the synthesis of NPs is illustrated in Figure 3B. As the concentration of the extract increased, the absorbance increased, suggesting a higher production of AgNPs. This can be attributed to the secondary reduction of Ag^+^ ions [32]. A greater quantity of the reducing agent facilitates the nucleation process. A previous study showed that lower concentrations of extracts result in the synthesis of fewer AgNPs, with some particles exhibiting irregular and anisotropic nanostructures [33]. To optimize the yield of synthesized AgNPs, 10 mg/mL EGb extract was selected for subsequent experiments.

#### 2.2.3. Effect of the Reaction pH

The size, deposition location, and stability of NPs are controlled by the solution pH [34]. Consequently, changes in the pH significantly affect the shape and position of the SPR peaks. The variations in the UV spectra under different pH conditions are illustrated in Figure 3C. A pH of 9 is optimal for obtaining the strongest SPR absorption band with symmetrical peaks, indicating that acidic conditions are unfavorable for AgNP synthesis. The pH of the solution primarily affects the surface charge of phytochemical functional groups. Vivek et al. demonstrated that the rapid synthesis of AgNPs under alkaline conditions can be attributed to the ionization of phenolic groups in *A. squamosa* extracts [35]. Limited synthesis occurred under acidic conditions because of the electrostatic repulsion of anions; conversely, distinct absorption peaks can be observed under alkaline conditions. This can be attributed to the critical role the acid–base environment plays in balancing the nucleation and growth processes of NPs [36]. Small spherical particles are obtained at high pH values. However, in certain experimental scenarios described in the literature, higher pH levels have resulted in size growth, particle aggregation, and reduced synthesis rates of AgNPs [37]. In addition, pH adjustments revealed that pH can affect the reaction rate and that the pH is directly proportional to the rate of color development in the reaction mixture. Furthermore, depending on the complexity of the plant species and composition, AgNPs are often synthesized within different pH ranges. Nahar et al. indicated that a large quantity of particles can be generated at pH 11 using *C. sinensis* peel extract [38].

#### 2.2.4. Effect of the Reaction Temperature

The effect of temperature on the synthesis of AgNPs is shown in Figure 3D. The results indicate the temperature-dependent synthesis of AgNPs, consistent with the findings reported in the literature. Under identical conditions, a temperature of 90 °C yields a more intense SPR peak than synthesis at room temperature, suggesting the generation of a larger amount of AgNPs. At temperatures exceeding 60 °C, a sharp SPR peak emerged and shifted towards shorter wavelengths, known as a blue shift. Furthermore, the SPR peaks observed at higher temperatures are asymmetric [39]. This phenomenon implies that AgNPs produced in these environments are not homogeneously dispersed despite the tendency towards a reduced particle size. Accordingly, a temperature of 60 °C was used to synthesize AgNPs. The temperature influences the nucleation of NPs. Although some researchers have opted to synthesize AgNPs from plant sources at room temperature, Tippayawat et al. demonstrated that a longer time is required to obtain AgNPs at lower temperatures [40]. Additionally, increasing the temperature enhances the kinetics of ionic reactions, thereby expediting the consumption of Ag^+^ ions and their conversion to Ag atoms.

#### 2.2.5. Effect of the Reaction Time

The results shown in Figure 3E demonstrate that the synthesis of AgNPs is completed within 45 min. With increasing reaction time, the nucleation of NPs occurs at a slower rate. However, the shapes of the absorption peaks remain unchanged. In addition, the position and width of the SPR peaks also remain constant, indicating that the reaction time does not affect the size and morphology of AgNPs. However, the absorption intensity decreases when the reaction time is extended to 60 min. This decline is attributed to the aggregation and instability of nanosilver [41]. According to Mittal et al., maximum absorption can be observed after 12 h, and particle aggregation occurs after 24 h in AgNPs synthesized using *Syzygium cumini* fruit extracts [42]. Therefore, enhancing the efficiency of biosynthesis compared to chemical synthesis is crucial. Based on the use of Gb for the synthesis of AgNPs, the time expenditure can be minimized to within 45 min.

### 2.3. Characterization of the Synthesized AgNPs

#### 2.3.1. TEM Analysis

The initial synthesis of AgNPs was confirmed using UV spectrophotometry. Under optimized conditions, TEM enables the visual determination of the morphology and size of the AgNPs. At low magnification (Figure 4), dispersed and non-agglomerated AgNPs can be observed. Consistent with previous findings, most AgNPs synthesized from leaf extracts exhibit a spherical shape. The particle size histogram displays an average AgNP size ranging from ~8 to 9 nm. Consistent with this investigation, the size range of AgNPs synthesized from algae, bryophytes, ferns, and gymnosperms is below 50 nm [43]. Furthermore, at high magnification, the encapsulation of AgNPs by plant envelopes stabilizes NPs with a thin organic material. The deposition and growth of bio-organic phase crystals during synthesis contribute to the linear pattern in the assembly of AgNPs.

#### 2.3.2. XRD Analysis

XRD analysis was employed to assess the crystalline structures of AgNPs derived from EGb (Figure 5). Noticeable diffraction peaks were observed within the 2θ range of 20°–80°. Specifically, the peaks located at 38.26°, 46.20°, 64.74°, and 77.06° correspond to the four characteristic diffraction peaks of Ag: the (111), (200), (220), and (311) planes, respectively. These phase planes are the primary contributors to the synthesis of AgNPs, which adopt face-centered cubic crystalline structures. The peaks observed at 55° and 57° belong to AgCl and are consistent with previous studies [44]. Moreover, aqueous plant extracts may simultaneously produce Ag and AgCl. Based on calculations using Debye–Scherrer’s formula, the average crystal size of the AgNPs was estimated to be approximately 6 nm.

#### 2.3.3. FTIR Analysis

FTIR analysis reveals the molecular vibrations of a sample in specific infrared regions (corresponding to the twisting, bending, and stretching of chemical bonds) to identify phytochemicals involved in AgNP coating [45]. Figure 6 shows the peak transmittance of EGb and AgNPs. The FTIR spectra of nanosilver synthesized from plant extracts exhibit significant and minor shifts. The peak at 3434.11 cm^−1^ in the nanosilver spectrum, attributed to the O-H or N-H stretching of phenolics in the plant extracts, shifted to 3429.10 cm^−1^. The characteristic bands at 2918.24 cm^−1^ and 2848.77 cm^−1^ are attributed to the C-H stretching vibrations. The band observed at 1614.84 cm^−1^ corresponds to the amide band, which is generated by the stretching of C=O in amides I, II, and III [46]. This may be due to the vibration of protein amides, which become more prominent in the AgNPs spectrum [47]. The presence of carbon chain compounds, such as aliphatic compounds [48], can be inferred from the bands at 1039.22, 2918.24, and 2848.77 cm^−1^. The band at 1373.35 cm^−1^ in the AgNP spectrum corresponds to the aromatic ring skeleton, possibly corresponding to compounds such as flavonoids and phenols in the leaf extracts [26]. Moreover, FTIR analysis demonstrated that EGb-capped AgNPs have spectra highly similar to EGb. This indicates that the structure of EGb has not been disrupted but rather coated on the surface of AgNPs, maintaining their stability. All these vibrational bands evidence the successful encapsulation of polyphenols, terpenoids, proteins, and polysaccharides from plants onto the NPs. In agreement with the results of the FTIR analysis, LC-MS analysis demonstrated the presence of flavonoids and phenolic acids in the leaves of Gb plants. Plant secondary metabolites are crucial for AgNP synthesis.

### 2.4. Analysis of Potential Antimicrobial Activity

#### 2.4.1. MBC and MIC Measurements

The broth dilution method was used to determine the MIC and MBC of AgNPs against the four bacterial species. Briefly, bacteria were co-cultured with various concentrations of AgNPs for 24 h, and the MIC was identified as the lowest concentration in a clear tube at which no bacterial growth was observed. The MIC of AgNPs on *E. coli* was 4 μg/mL, whereas the MIC on *S. aureus*, *P. aeruginosa*, and *A. baumannii* was 8 μg/mL (Figure 7 and Table 3). A reduced number of individual *E. coli* colonies was observed at an AgNP concentration of 4 μg/mL, whereas no colony growth was detected at 8 μg/mL, indicating the MBC. The MBC value for *S. aureus*, *P. aeruginosa*, and *A. baumannii* is 32 μg/mL. These results suggest that *E. coli* exhibits a greater sensitivity to AgNPs, which can be attributed to its structure. Agnihotri et al. noted an abundance of AgNPs deeper within the cell membrane surface of *E. coli* than within the intracellular space, suggesting that the primary AgNP target site in *E. coli* is the cell membrane [49]. The depletion of metals disrupts the cellular barrier, modifies membrane permeability, and facilitates the liberation of Ag^+^ ions towards the cytoplasmic membrane. In contrast, the thickened peptidoglycan layer and the presence of efflux transporters in the cell wall of the Gram-positive *S. aureus* are believed to play a critical role in hindering or delaying the invasion of NPs, thereby protecting against cell death [50]. However, it is important to note that the sensitivity of bacteria to AgNPs cannot be solely assessed based on variations in the cell membrane composition [49]. Rodríguez-León et al. indicated that AgNPs synthesized using *Rumex hymenosepalus* extract exhibit a more potent bacteriostatic effect against *S. aureus* than against *E. coli* [51]. The involvement of plant metabolites further contributes to the disruption of cell membranes, synergizing with AgNPs to generate robust antimicrobial capacity [9].

AgNPs also demonstrate a range of physical properties resulting from diverse synthesis methods. The antibacterial effects of AgNPs can be influenced by their morphology, size, dispersibility, and the choice of capping agents [44]. Several mechanisms have been identified as possible explanations for the antibacterial action of AgNPs. Figure 8 represents the potential mechanism of antimicrobial activity of AgNPs. Firstly, AgNPs adhere to and accumulate on the cell wall and membrane, leading to the formation of pits and causing the disruption and degradation of the membrane. Secondly, AgNPs interact with respiratory chain proteins and transport proteins on the cell membrane, interfering with the transport of phosphate ions, thus affecting the respiratory chain and inhibiting ATP generation. Thirdly, upon penetration through the compromised cell membrane, AgNPs interact with intracellular biomolecules, inhibiting transcription, translation, and protein synthesis, disrupting carbohydrate metabolism, and inducing DNA denaturation. Consequently, these disruptions in cellular functions ultimately result in cell death [52]. The generation of reactive oxygen species (ROS) is also considered one of the mechanisms underlying the antimicrobial activity of AgNPs. AgNPs and Ag^+^ contribute to the production of hydroxyl radicals and ROS, thereby increasing the levels of cellular ROS and inhibiting bacterial cell growth [53]. The denaturation of DNA is also associated with ROS production, as oxidative damage to amino acids caused by oxygen radicals associated with AgNPs and Ag^+^ ion exposure can disrupt the protein structure [54]. Additionally, the choice of capping agents can influence the release of AgNPs. Lipid-stabilized AgNPs exhibit better dispersibility compared to uncoated AgNPs and also demonstrate enhanced Ag^+^ ion release capability and antibacterial efficacy [55].

#### 2.4.2. Growth Kinetics Assay

To gain a deeper understanding of the interaction between nanomaterials and microorganisms, the growth curves of bacteria were assessed using a microplate reader during treatment with various concentrations of AgNPs over a 24 h period. The optical density (absorbance) was measured at 600 nm (Figure 9). The results agree with the MIC and MBC values. The bacteriostatic effect of AgNPs exhibits a concentration-dependent relationship, with higher concentrations of AgNPs showing stronger inhibition of bacterial growth. Several researchers have reported the dose-dependent antimicrobial effects of AgNPs synthesized using plant extracts [56]. Furthermore, we observed that the growth of *S. aureus*, *P. aeruginosa*, and *A. baumannii* was significantly inhibited, but possibly not completely eliminated, by treatment with AgNPs at a concentration of 16 μg/mL, consistent with the single-colony growth observed in the MBC results. At the MBC, the growth of the four bacteria was not detected over 24 h.

#### 2.4.3. Agar Well Diffusion Assay

The antibacterial activities of AgNPs at various concentrations against the four tested bacteria were investigated using the agar well diffusion method. In addition, inhibition zones were also evaluated for EGb, standard antibiotics, and AgNO_3_, as shown in Figure 10 and Table 4. The diameters of the inhibition zones were measured in triplicate. EGb does not inhibit the growth of the tested bacteria. As expected, the antimicrobial effect of AgNO_3_ as a control is greater than that of AgNPs. AgNO_3_ can rapidly disperse many Ag^+^ ions, showing antibacterial activity [30]. On the other hand, AgNPs are active substances that can release silver ions in a controlled and prolonged manner, making them more suitable for combating chronic infections caused by microorganisms. The low biocompatibility and functionality of AgNO_3_ must be distinguished from the direction of application of AgNPs. As the AgNP concentration increases, the zone of bacterial growth inhibition on the solid medium also increases. Even at low AgNP concentrations (8 μg/mL), a clear inhibition zone began to appear for the four bacteria. At high AgNP concentrations (32 μg/mL), the inhibition zones for *E. coli*, *S. aureus*, *P. aeruginosa*, and *A. baumannii* were measured to be 12.62, 14.51, 16.33, and 13.35 mm, respectively. These results indicate that AgNPs with diameters ranging from 8 to 9 nm synthesized using EGb extracts inhibited the growth of the tested bacteria, even at low concentrations.

The bacterial cell wall may hinder the uptake of larger AgNPs, whereas AgNPs smaller than 10 nm exhibit the best antibacterial activity because of their increased specific surface area [57]. The mechanism of microbial killing within this size range may involve both intracellular release and direct contact. In addition, the dissolution kinetics increase with decreasing NP size, thereby enhancing antibacterial potential through a non-monolithic mode of action [50]. *E. coli* does not exhibit the same sensitivity using this method as that observed in the MIC test. This discrepancy may be attributed to differences in the testing methods because the agar well diffusion approach may lack specificity compared with mixed cultures using bulk AgNPs. Therefore, exposure to bacteria in mixed cultures may provide a more accurate assessment. Furthermore, *P. aeruginosa* was the most sensitive to AgNPs and formed the largest inhibition zone compared with other bacteria, which is consistent with the findings by Jeeva et al. [58]. AgNPs exhibit strong antibacterial effects, particularly against bacterial biofilms. A study has shown that AgNPs at a concentration four times the MIC significantly inhibited the formation of biofilms in all strains of *S. aureus* [59]. Another study has identified the molecular targets of silver in *S. aureus*, demonstrating that Ag^+^ can interact with 38 proteins in *S. aureus* through various pathways such as glycolysis, the oxidative branch of the pentose phosphate pathway, and the ROS-mediated stress defense system, exerting bactericidal effects [60]. This multi-target mode of action also contributes to the sustained antimicrobial efficacy of AgNPs. Adebayo-Tayo et al. tested the antibacterial activity of 10 nm diameter AgNPs synthesized using the methanol extract of *Oscillatoria* sp. against seven clinically pathogenic bacteria [61]. Their results showed that AgNPs exhibited the highest biofilm inhibition activity against *P. aeruginosa*, which may be associated with amyloidosis. FapC, a component of the extracellular amyloid network of *P. aeruginosa*, can enhance its antibiotic resistance, leading to infections that are challenging to treat. Another study also demonstrated that low concentrations of AgNPs can disrupt bacterial biofilms by inhibiting FapC fibrillation [62]. Additionally, the primary mechanism of the antibacterial effect of a colloidal nanosilver formulation called Silversol against *P. aeruginosa* involved the disruption of iron homeostasis and induction of nitrosative stress. Importantly, Silversol selectively targeted proteins in *P. aeruginosa* without affecting beneficial bacteria found in humans, especially the human gut, making it a potential candidate for antibiotic development [63]. Moreover, some studies have investigated the synergistic antibacterial effects of combining nanosilver and antibiotics to mitigate the development of bacterial resistance. Twyman-stabilized AgNPs could decrease the MIC of gentamicin [64]. Additionally, combining AgNPs with polymyxin B or levofloxacin has exhibited effective synergistic action against clinically isolated carbapenem-resistant strains of *A. baumannii* [65]. Furthermore, a study has shown that 10 nm AgNPs showed the highest inhibition of *P. aeruginosa* biofilms. AgNPs have also been shown to enhance the interaction between aztreonam and bacteria by penetrating the biofilm matrix [66]. These reports suggest that reducing the particle size is advantageous in enhancing both the antibacterial efficacy and availability of AgNPs. This low-dose combination therapy has the potential to mitigate the adverse effects of antibiotics. Collectively, these data suggest that our synthesized AgNPs, with their narrow size characteristics, hold promise for the treatment of bacterial infections, either as standalone agents or in combination therapies.

### 2.5. Cytotoxicity

Nanosilver has been identified as a potential hazard, so its toxicity to normal cells must be assessed. Previous studies have shown that the mammalian cell toxicity induced by AgNPs is influenced by both the physical properties of the NPs and the cell types involved [67]. Therefore, we used various cell lines to evaluate the toxicity of AgNPs. The cytotoxicity of AgNPs synthesized in this study was evaluated using L929 cells, rBMSCs, and hPDLCs. The cell survival percentages at different AgNP concentrations after 24 h of treatment are shown in Figure 11. The results indicate that AgNPs do not affect the growth of normal cells from three different sources at concentrations below 16 μg/mL. However, L929 and hPDLC cells showed apparent cytotoxicity at 64 μg/mL. rBMSCs are the most resistant to nanosilver. Furthermore, the IC50 value for L929 cells was 22.33 μg/mL, indicating that half of these cells would be dead at this concentration. Interestingly, cell growth was promoted at low AgNP concentrations in all three normal cell lines, although cytotoxicity was observed at AgNP concentrations exceeding 16 mg/mL.

There is evidence that cytotoxicity increases with the size and dose of AgNPs [68]. Xu et al. showed that AgNPs with a diameter of 10 nm exhibit lower cytotoxicity than AgNPs with a diameter of 40 nm [69]. However, it has been reported that small-diameter AgNPs can upregulate apoptosis because of their rapid internalization [70]. This variability observed among particles used in different studies may be related to the specific effects of AgNPs on different cells. Depending on their size and the species of plant extract used, AgNPs can penetrate cells via diffusion, phagocytosis, endocytosis, and active transport [71]. Mammalian cells protected by complex cell membranes tend to show increased resistance to AgNPs, which allows primary cells to exhibit lower cytotoxicity at MBC values [72]. Our results are consistent with these findings. In addition, surface-adsorbed plant extracts contain secondary metabolites that synergize with NPs to enhance their biological functions. Hu et al. discovered that polyphenols from the *Bauhinia acuminate* plant flower extract act as nutrients and promote the proliferation of L929 cells at low AgNP concentrations [62]. However, the exact molecular targeting mechanism of AgNPs has not been determined, although most studies were focused on oxidative stress. The results of this study highlight the presence of ROS, primarily in mitochondria, which disrupts ATP synthesis and leads to DNA damage, lipid peroxidation, and protein carbonylation. DNA damage is mainly manifested as single-strand breaks and increased DNA tail length. Furthermore, the activation of the caspase cascade plays a critical role in triggering apoptosis [73]. In addition, the same concentration of AgNPs demonstrated effective bacterial inhibition and cytocompatibility. This result is not contradictory because Sanyasi et al. previously reported that prokaryotic membrane proteins have fewer exposed functional thiol groups than eukaryotic proteins, facilitating bacterial binding and damage induced by AgNPs [74].

## 3. Materials and Methods

### 3.1. Obtaining Ginkgo biloba Leaf Extract

Fresh ginkgo leaves were collected from ginkgo trees on campus (Northwestern University, Xi’an, China) and washed with deionized water to remove any dust or impurities. The leaves were then dehydrated in an oven at 40 °C for 48 h. One gram of dried leaves was cut into pieces and mixed with 100 mL of deionized water. The mixture was heated at 90 °C for 60 min and filtered using Whatman No. 1 filter paper. The filtered extract solutions were sealed and stored in a refrigerator at 4 °C for future use.

### 3.2. High-performance Liquid Chromatography Coupled with Mass Spectrometry (HPLC-MS) Analysis

Take 1 mL of the filtrate diluted 10-fold, filter it through a 0.22 μm membrane, and transfer it to a liquid-phase bottle. The sample separation was performed using the ACQUITY ultra-high-performance liquid chromatography system (Waters, Milford, MA, USA) and the Waters ACQUITY HSS C18 chromatographic column (2.1 × 100 mm, 1.8 μm). The mobile phase A is 0.1% formic acid, and mobile phase B is pure methanol. The injection volume is 3.0 μL, and the column temperature is 35 °C. The eluate from the chromatographic system was ionized using the Agilent 5600 time-of-flight high-resolution mass spectrometry equipped with an electrospray ion source in negative ion mode to obtain mass spectrometry data. The operating parameters are as follows: the ion source temperature is 500 °C; the capillary voltage is 4500 V; the nebulizer gas pressure is 50 psi; collision voltage is −10 eV; the mass scan range is 50–1000 *m*/*z*.

### 3.3. Synthesis of AgNPs

To synthesize AgNPs, we combined 5 mL of the plant extracts with 5 mL silver nitrate (AgNO_3_, Macklin, Shanghai, China) in a 1:1 ratio. The pH of the mixture was adjusted by adding 0.1 M sodium hydroxide (Macklin, Shanghai, China) or 0.1 M hydrochloric acid (Macklin, Shanghai, China). The mixture was heated in a blender to facilitate the reaction. All reactions were conducted in a dark environment to prevent the photoactivation of the mixture.

To investigate the effects of different reaction conditions on the synthesis of AgNPs, various parameters, including the pH, reaction temperature, reaction duration, and concentrations of ginkgo extract and AgNO_3_, were evaluated. To evaluate the effect of AgNO_3_ concentration, the plant extracts were mixed with different concentrations of AgNO_3_ (2, 4, 6, 8, and 10 mM) at a pH of 8 and a temperature of 40 °C. The mixture was stirred for 30 min. Subsequently, under the same conditions, the optimal concentration of AgNO_3_ was mixed with different concentrations of the plant leaf extracts (2, 4, 6, 8, and 10 mg/mL). The optimal conditions were then employed for subsequent experiments. To examine the effect of pH, the reaction pH was adjusted to 3, 5, 7, 9, or 11. To analyze the effect of the reaction temperature, the mixture was heated at 30, 40, 60, 80, and 90 °C during the reaction. To assess the effect of the reaction duration, the following reaction times were chosen: 30, 45, 60, 75, and 90 min. Each experiment was carried out at least in triplicate.

### 3.4. Characterization of Biosynthesized AgNPs

#### 3.4.1. Ultraviolet–Visible Spectroscopy

A UV spectrophotometer (Shimadzu UV-2500, Kyoto, Japan) was used to confirm the synthesis of AgNPs. UV spectrophotometry allows us to observe the absorption spectra of the reactants in the 300–700 nm range at a resolution of 0.5 nm. A quartz cuvette with an optical range of 10 mm was used for these measurements. Deionized water was utilized as the reference sample for the calibration.

#### 3.4.2. Transmission Electron Microscopy (TEM)

TEM (JEOIF-200, Tokyo, Japan) was used to analyze the morphology and size of the AgNPs. Liquid samples were dried by depositing drops onto a copper mesh coated with a carbon layer. Representative images of AgNPs were captured and examined.

#### 3.4.3. X-ray Diffraction (XRD)

XRD was used to analyze the crystal structure of the AgNPs. The X-ray diffractometer (SmartLab SE, Tokyo, Japan) was operated using Cu-Kα radiation at 40 kV and a current of 50 mA. The scanning range and rate were set to 30° to 80° and 2°/min, respectively. Crystal size was estimated using Debye–Scherrer’s formula:(1)$$D=\frac{K\;\lambda }{B\;COS\theta }$$

#### 3.4.4. Fourier Transform Infrared Spectroscopy (FTIR)

The AgNP powders were analyzed using FTIR (TENSOR27, Bruker, Germany) to determine the presence of biological elements from the plant extracts responsible for reducing and capping silver ions during synthesis. Briefly, the dried sample powder was finely ground and subjected to FTIR analysis with a resolution of 1 cm^−1^ in the spectral range of 4000–450 cm^−1^.

### 3.5. Analysis of Potential Antimicrobial Activity

#### 3.5.1. Bacterial Cell Culture

*Staphylococcus aureus* (ATCC 25923), *Escherichia coli* (ATCC 25922), *Acinetobacter baumannii* (ATCC 17978), and *Pseudomonas aeruginosa* (PAO1) were used to assess the antimicrobial activity of the AgNPs. Bacteria were streaked on Luria–Bertani (LB) plates to restore viability and then incubated at 37 °C for 12 h. Individual colonies of each strain were selected and inoculated into LB medium for further growth at 37 °C for 24 h. After that, the turbidity of the bacterial cultures was adjusted to 0.5 McFarland.

#### 3.5.2. Determining the Minimum Inhibitory Concentration (MIC) and Minimum Bactericidal Concentration (MBC)

The MIC and MBC of AgNPs synthesized from Gb were determined using the broth dilution method to assess their antimicrobial efficacy against four bacterial species. Briefly, AgNPs were serially diluted in sterile water to obtain different concentrations. Then, 2 mL of each dilution was mixed with 2 mL of a 1 × 10^6^ CFU/mL bacterial solution in separate tubes. The concentrations of the AgNPs in the tubes were 64, 32, 16, 8, 4, 2, 1, and 0.5 μg/mL. LB broth containing the bacterial strains served as the positive control, while gentamicin and ampicillin were used as negative controls. All tubes were incubated at 37 °C and 250 rpm for 24 h. The highest dilution at which no visible microbial growth was observed was determined as the MIC. To determine the MBC of the AgNPs, 100 μL of bacterial suspension from each clear tube was applied to nutrient agar plates and incubated for 24 h. The lowest concentration of AgNPs without colony growth on the agar was considered the MBC. Each experiment was carried out at least in triplicate.

#### 3.5.3. Growth Kinetics Assay

Sterile 96-well plates were utilized, and two-fold AgNP dilutions (650, 320, 160, 80, 40, 20, 10, 5 μg/mL) were prepared using sterile water. Each bacterial suspension was diluted to 1 × 10^6^ CFU/mL, and 90 μL of this suspension was added to each well of the well plate, followed by the sequential addition of 10 μL of the AgNPs. Positive control wells contained 10 μL of liquid LB medium, while negative control wells contained antibiotics. Subsequently, the plate was placed in a multifunctional enzyme marker (SynergyH1, VT, USA), and the optical density (OD)_600_ was continuously measured for 24 h.

#### 3.5.4. Agar Well Diffusion Assay

The agar well diffusion method was used to assess the antibacterial activity of AgNPs at varying concentrations [75]. Briefly, 100 µL of each bacterial suspension (1 × 10^6^ CFU/mL) were evenly spread on an agar plate. A gel puncture was performed to create 5 mm diameter wells on the plate. Subsequently, different concentrations of AgNPs, plant extracts, silver nitrate solution, and antibiotics were sequentially added to the wells of the agar plates. The presence of a transparent area around the well was recorded as an inhibition of bacterial growth. The diameter of the zone of inhibition (mm) was measured after incubation at 37 °C for 24 h. Each experiment was carried out at least in triplicate.

### 3.6. Cytotoxicity Assay

#### 3.6.1. Cell Culture

The cytotoxicity of the AgNPs was assessed using murine L929 fibroblasts, rat bone mesenchymal stem cells (rBMSCs), and human periodontal ligament cells (hPDLCs). L929 fibroblasts were purchased from Procell Life Science & Technology Co., Ltd. (Wuhan, China), and hPDLCs were a generous gift from Dr. Zhenhua Yang, School of Stomatology of Fourth Military Medical University. L929 cells and hPDLCs were cultured in Dulbecco’s Modified Eagle Medium (DMEM; Gibco, MA, USA), whereas rBMSCs were cultured in Minimum Essential Medium-Alpha (α-MEM; Gibco, MA, USA). Both media were supplemented with 10% fetal bovine serum (FBS, Sijiqing Bioengineering, Hangzhou, China) and 1% penicillin–streptomycin (Beyotime Biotechnology, Shanghai, China).

Primary rBMSCs were isolated from one-week-old Sprague–Dawley rats by isolating the femur and tibia and removing the surrounding muscle and connective tissue using sterilized forceps and scissors. After rinsing three times with a phosphate buffer (PBS; Beyotime Biotechnology, Shanghai, China), the bones were placed in α-MEM containing 10% FBS. Finally, the femurs were cut to expose the metaphyseal cavity. The bone marrow cavity was flushed with α-MEM using a 1 mL syringe to obtain a cell suspension. These cells were transferred to culture flasks and maintained at 37 °C with 5% CO_2_ for subsequent experiments and analyses.

#### 3.6.2. Cytotoxicity Assays

The cytotoxicity of AgNPs on L929 cells, rBMSCs, and hPDLCs was assessed using the CCK-8 (Beyotime Biotechnology, Shanghai, China) assay. Briefly, 1 × 10^4^ cells were seeded per well of a 96-well plate and incubated for 12 h at 37 °C. AgNPs were then added to the cells to obtain final concentrations of 64, 32, 16, 8, 4, 2, 1, and 0.5 μg/mL. After 24 h of culture, the spent medium was replaced with fresh medium containing 5% CCK-8 solution, and the cells were incubated for another 2 h. The absorbance of each well at 450 nm was measured using a multifunctional enzyme marker (SynergyH1, VT, USA). Each experiment was carried out at least in triplicate.

### 3.7. Statistical Analyses

All data are expressed as the mean ± standard deviation (SD). Data analysis and graphing were conducted using the Origin 2021 and GraphPad Prism 8.0.1 software. Multiple groups were statistically analyzed using one-way analysis of variance and Student–Newman–Keuls post hoc tests. Probability values < 0.05 were considered significant. All experiments were repeated at least three times.

## 4. Conclusions

Nanosilver produced via phytosynthesis has a wide range of chemical compositions and offers promising prospects in biomedical applications. The narrow structure of the synthesized nanosilver effectively inhibited the growth of microorganisms. In this investigation, we employed an environmentally friendly and cost-effective approach by synthesizing AgNPs using an aqueous extract derived from Gb leaves. We also employed various methods to assess the morphology and antimicrobial effects of the NPs. Our results demonstrate that EGb, enriched with various phytochemicals, effectively served as a capping and reducing agent, facilitating the synthesis of stable and small-sized AgNPs. TEM and XRD results showed that the average size of the AgNPs was <10 nm. These NPs also substantially inhibited the growth of four pathogenic bacterial strains at MIC values. Furthermore, the optimal concentration of 16 μg/mL AgNPs was identified between the MIC and MBC values. This concentration achieved a delicate balance between antibacterial effectiveness and cell viability. Overall, the findings of this study highlight the optimized conditions to facilitate the synthesis of small-sized AgNPs using Gb, aligning with the principles of green synthesis while offering potential from sustainable sources. The precisely controlled dispersion, thorough characterization, excellent antibacterial activity, and cell viability observed in these AgNPs provide valuable insights for future applications.

## Figures and Tables

**Figure 1 ijms-25-01913-f001:**
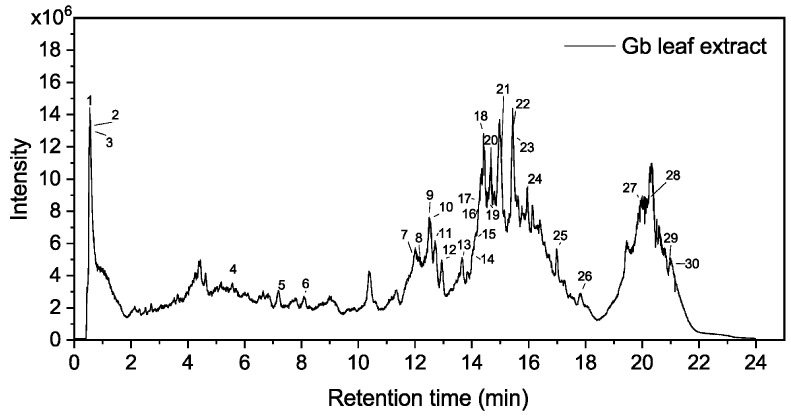
Gb leaf extract sample total ion chromatogram.

**Figure 2 ijms-25-01913-f002:**
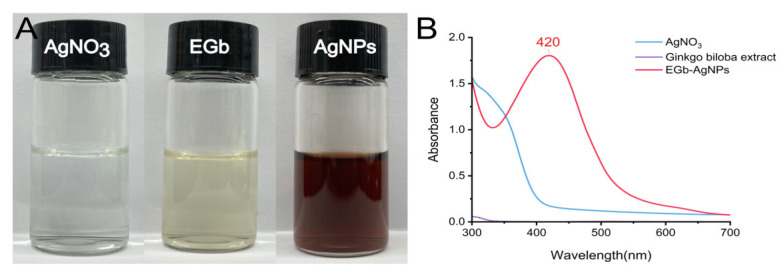
Color change of the reaction mixture (**A**) and UV spectroscopy absorption spectra of AgNPs successfully synthesized using plant extracts (**B**). The peak of AgNPs appears at 420 nm.

**Figure 3 ijms-25-01913-f003:**
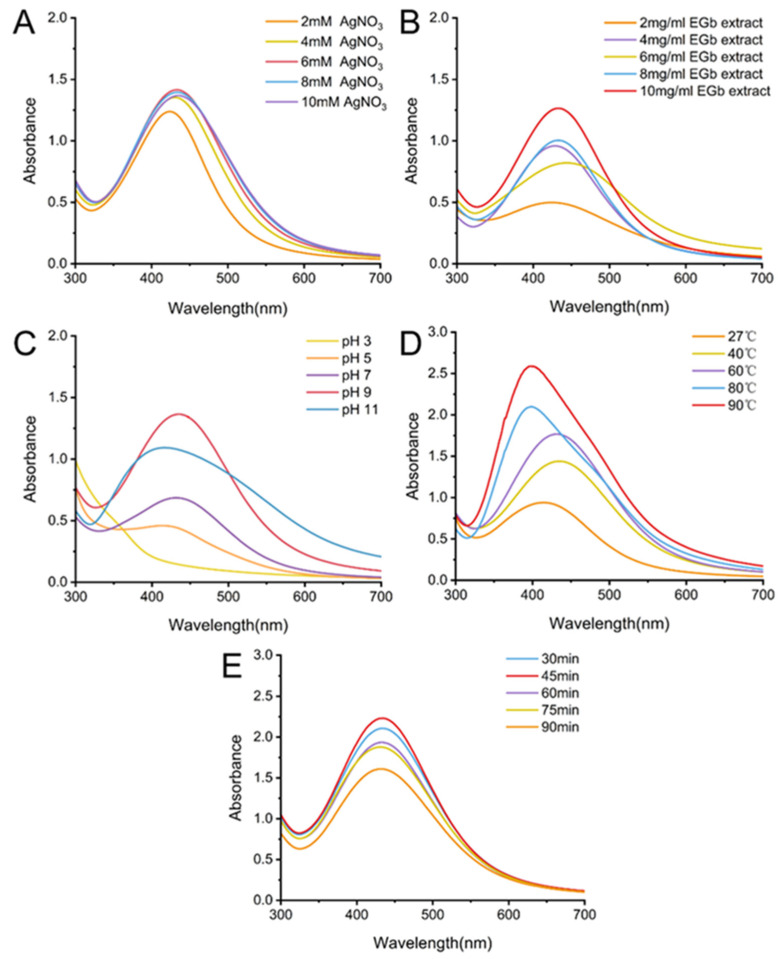
Optimization of the synthesis of AgNPs at different parameters: (**A**) EGb concentration, (**B**) AgNO_3_ concentration, (**C**) pH, (**D**) reaction temperature, and (**E**) reaction duration.

**Figure 4 ijms-25-01913-f004:**
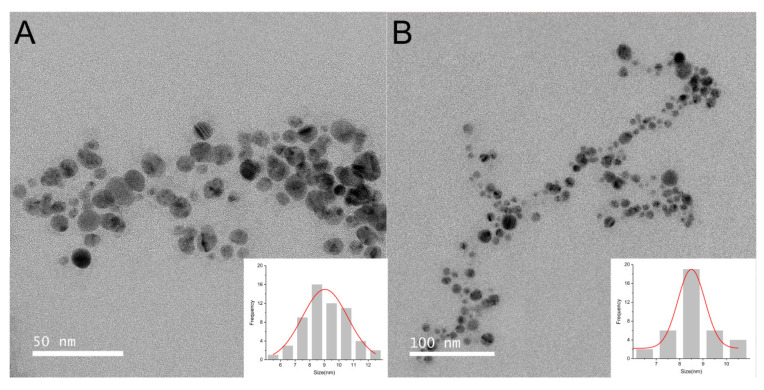
TEM analysis of AgNPs. TEM images of AgNPs at 50 nm (**A**) and 100 nm (**B**). Insets show the size distribution of the synthesized AgNPs.

**Figure 5 ijms-25-01913-f005:**
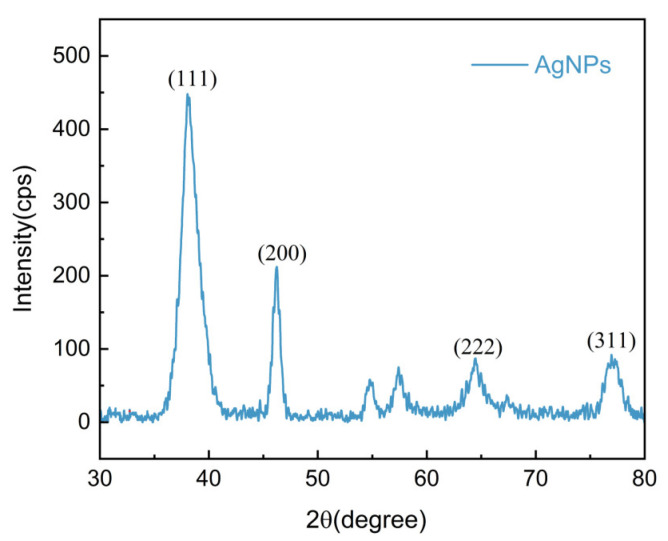
XRD pattern of AgNPs. The (111), (200), (220), and (311) planes belong to AgNPs.

**Figure 6 ijms-25-01913-f006:**
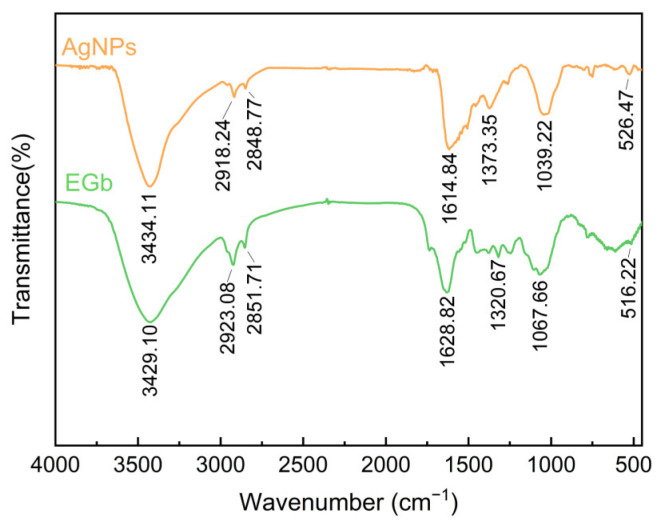
FTIR spectra of the EGb and AgNPs.

**Figure 7 ijms-25-01913-f007:**
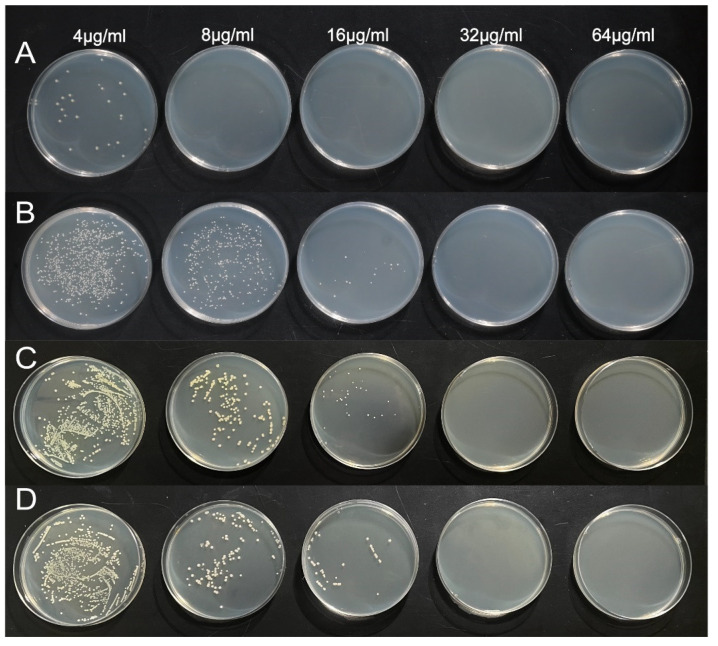
Colony growth under different concentrations of AgNPs. (**A**) *E. coli*, (**B**) *S. aureus*, (**C**) *P. aeruginosa*, and (**D**) *A. baumannii*.

**Figure 8 ijms-25-01913-f008:**
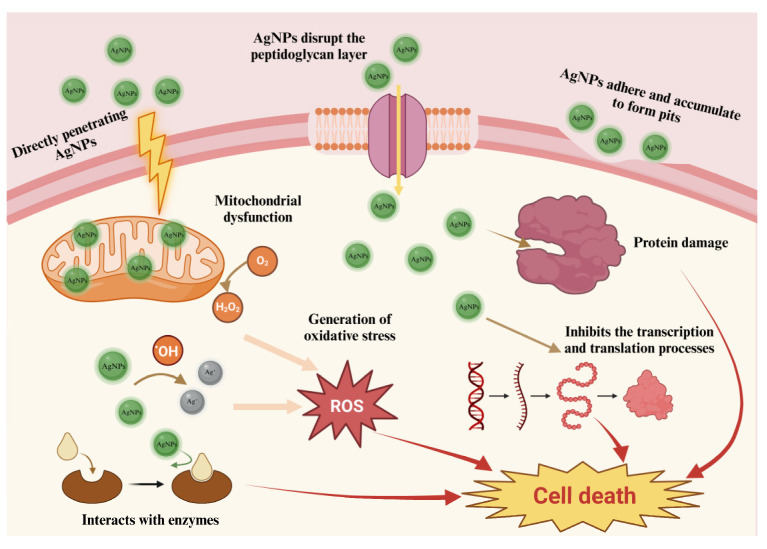
The probable mechanism of the antibacterial activity of AgNPs.

**Figure 9 ijms-25-01913-f009:**
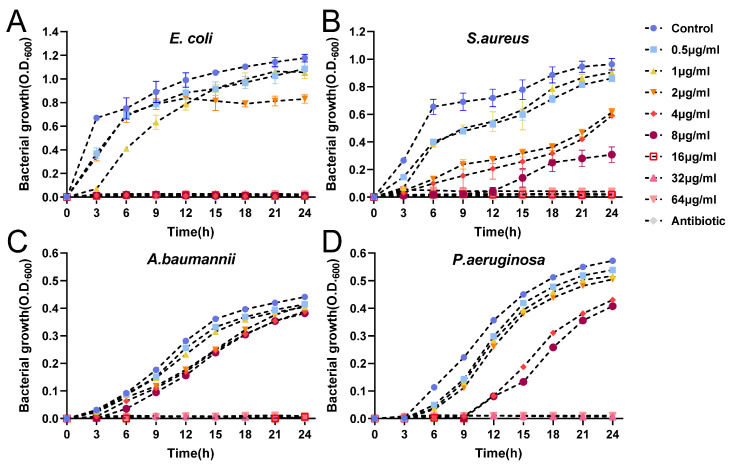
Growth curves of bacteria under different concentrations of AgNPs. (**A**) *E. coli*, (**B**) *S. aureus*, (**C**) *P. aeruginosa*, and (**D**) *A. baumannii*.

**Figure 10 ijms-25-01913-f010:**
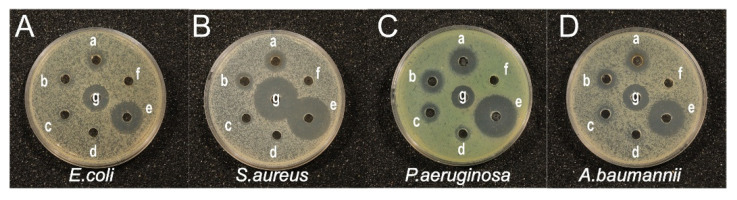
Antibacterial activities of AgNPs at different concentrations. (**A**) *E. coli*, (**B**) *S. aureus*, (**C**) *A. baumannii*, and (**D**) *P. aeruginosa.* a–d: 32, 16, 8, and 4 μg/mL of AgNPs, respectively; e, 1 mM AgNO_3_; f, EGb; g, antibiotics.

**Figure 11 ijms-25-01913-f011:**
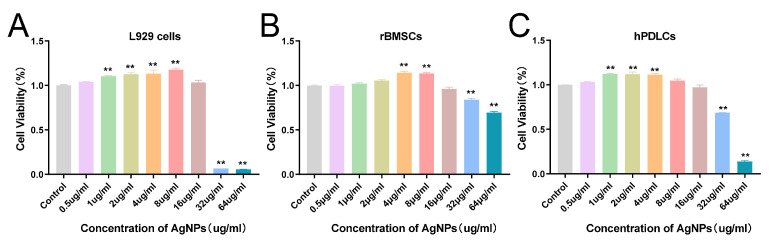
Cytotoxicity studies of AgNPs against L929 cells (**A**), rat bone mesenchymal stem cells (**B**), and human periodontal ligament cells (**C**). Significant changes are shown. ** *p* < 0.01.

**Table 1 ijms-25-01913-t001:** Different compounds identified from the leaf extracts of Gb via HPLC-MS analysis.

Peak No.	RT (min)	*m*/*z*	Adduct/Charge	Main Fragments Ions	Compound Identification	Compound Class
18	14.41	593.1478	[M-H]^−^	284,593	Kaempferol-3-O-rutinoside isomer	Flavonoid
27	19.94	577.2663	[M-H]^−^	225, 577	Chrysin 7-gentiobioside	Flavonoid
20	14.65	623.1584	[M+FA-H]^−^	316, 470, 621	Isorhamnetin-3-O-rutinoside	Flavonoid
3	0.57	191.0561	[M-H]^−^	59, 85, 127, 191	Quinic acid	Phenolic acid
11	12.71	609.1445	[M-H]^−^	300, 609	Quercetin-3-O-rutinoside	Flavonoid
22	15.45	739.184	[M-H]^−^	284, 593, 739	Tetrahydroxyflavones rutinosyl-rutinoside	Flavonoid
10	12.52	739.2055	[M-H]^−^	284, 575, 739	Kaempferol 3-O-(2,6-di-O-alpha-L-rhamnopyranosyl)-beta-D-galactopyranoside	Flavonoid
30	21.02	577.2661	[M-H]^−^	80, 299, 575	Chrysin 7-gentiobioside	Flavonoid
21	15.03	593.1476	[M-H]^−^	284, 413, 593	Kaempferol-rhamnosyl-glucoside	Flavonoid
12	12.94	769.2158	[M-H]^−^	314, 769	Typhaneoside	Flavonoid
1	0.53	377.0841	[M+Cl]^−^	113, 341	Sucrose	Carbohydrates
26	17.84	537.0804	[M-H]^−^	159, 375, 537	Amentoflavone	Flavonoid
24	15.96	739.1841	[M-H]^−^	284, 593	Robinin	Flavonoid
15	14.09	419.2265	[M-H_2_O-H]^−^	113	Swertimarin	Terpenoid
25	16.99	343.0812	[M-H]^−^	163, 313	Eupatilin	Flavonoid
16	14.23	447.091	[M-H]^−^	284, 447	Kaempferol-3-O-glucoside	Flavonoid
6	8.1	437.2364	[M-H]^−^	101, 391	Blinin	Terpenoid
8	12.19	463.0859	[M-H]^-^	151, 243, 303	Quercetin-3-O-galactoside	Flavonoid
9	12.47	463.086	[M-H]^−^	300, 445	Quercetin-3-O-galactoside	Flavonoid
17	14.25	609.1424	[M-H]^−^	300, 609	Aureusidin 4,6-diglucoside	Flavonoid
28	20.18	925.4212	[M-H]^−^	311, 925	Kaikasaponin III	Saponin
14	14.01	431.0958	[M-H]^−^	183, 255, 285	Genistin	Flavonoid
7	11.99	847.2623	[2M-H]^−^	423, 847	Ginkgolide B	Terpenoid
2	0.54	391.1033	[M-H]^−^	99, 217, 391	Hyodeoxycholic acid	acid
13	13.68	463.0851	[M-H]^−^	149, 301, 463	Hyperin	Flavonoid
5	7.18	439.1229	[M-H]^−^	97, 231, 365	Ginkgolide C	Terpenoid
29	20.94	293.2093	[M-H]^−^	79, 96, 167	Tetradecylsulfate	Fatty acids
19	14.54	477.1007	[M-H]^−^	314, 477	MQ-O-glucoside 3	Flavonoid
23	15.45	369.0888	[M-2H]_2_^−^	145, 255	Kaempferol 3-O-(2,6-di-O-alpha-L-rhamnopyranosyl)-beta-D-galactopyranoside	Flavonoid
4	5.58	451.2161	[M-H]^−^	59, 179, 241, 403	Calaliukiuenoside	Flavonoid

**Table 2 ijms-25-01913-t002:** Optimization conditions and UV results in AgNP synthesis.

Sample No.	Parameter	Condition	Variable	UV (nm)
1	Concentration of AgNO_3_ (mM)	Plant extract: 10 mg/mL pH: 9 T: 40 °C Time: 30 min	2	423
			4	430
			6	430
			8	434
			10	435
2	Concentration of plant extract (mg/mL)	AgNO_3_: 6 mM pH: 9 T: 40 °C Time: 30 min	2	420
			4	426
			6	440
			8	430
			10	430
3	pH	AgNO_3_: 6 mM Plant extract: 10 mg/mL T: 40 °C Time:30 min	3	No
			5	420
			7	433
			9	433
			11	405
4	Temperature (°C)	AgNO_3_: 6 mM Plant extract: 10 mg/mL pH: 9 Time: 30 min	27	413
			40	433
			60	430
			80	396
			90	396
5	Time (min)	AgNO_3_: 6 mM Plant extract: 10 mg/mL pH: 9 T: 60 °C	30	432
			45	431
			60	431
			75	428
			90	429

Final condition: AgNO_3_: 6 mM; plant extract: 10 mg/mL; pH: 9; T: 60 °C; time: 45 min.

**Table 3 ijms-25-01913-t003:** MBC and MIC values of *E. coli*, *S. aureus*, *P. aeruginosa*, and *A. baumanni*.

Microorganisms	AgNPs (μg/mL)								MIC (μg/mL of Ag)	MBC (μg/mL of Ag)
	0.5	1	2	4	8	16	32	64		
*E. coli*	+	+	+	−	−	−	−	−	4	8
*S. aureus*	+	+	+	+	−	−	−	−	8	32
*P. aeruginosa*	+	+	+	+	−	−	−	−	8	32
*A. baumanii*	+	+	+	+	−	−	−	−	8	32

+: turbidity; −: clarity.

**Table 4 ijms-25-01913-t004:** The size of the inhibition circle produced by AgNPs on the four test bacteria.

Zone Inhibition (mm)							
	AgNPs (μg/mL)						
Sample	4	8	16	32	AgNO_3_	Antibiotic	EGb
*E. coli*	7.03 ± 0.46	8.08 ± 0.35	11.29 ± 2.83	12.62 ± 1.17	21.11 ± 0.75	18.17 ± 0.33	NE
*S. aureus*	7.40 ± 0.15	9.01 ± 0.50	11.90 ± 0.41	14.51 ± 0.57	28.64 ± 0.37	29.73 ± 0.67	NE
*P. aeruginosa*	9.29 ± 0.96	11.02 ± 1.54	13.36 ± 1.74	16.33 ± 2.08	27.17 ± 0.81	14.68 ± 1.33	NE
*A. baumanii*	7.85 ± 0.27	8.50 ± 0.17	10.64 ± 0.90	13.35 ± 1.37	23.60 ± 1.58	16.94 ± 0.96	NE

NE: No inhibitory effect.

## Data Availability

The original contributions presented in this study are included in the article; further inquiries can be directed to the corresponding authors.

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
