# Peer review of "Green Synthesis of Narrow-Size Silver Nanoparticles Using Ginkgo biloba Leaves: Condition Optimization, Characterization, and Antibacterial and Cytotoxic Activities"

_ijms, 2024, doi:10.3390/ijms25031913_

Round 1
Reviewer 1 Report
Comments and Suggestions for Authors
The article of of Qi Ni et al described very interesting and voluminous research, in which the AgNPs synthesized and stabilized with the leaf extract of Ginkgo biloba were studied as antimicrobial agents. The work was performed with the use of wide range of research methods. The synthetic procedures, the characterization of obtained NPs, and antibacterial and cytotoxic properties of AgNPs is perfectly described. Thus, my questions to the authors have only debatable character and hand also touches on some experimental details.
1) In works related to the use of biological objects, the question always arises: will the composition of the extract change depending on the batch of the biological product and the place of origin, and how will this affect the final stage of use?
2) What is the pH of the extract solution and is it possible to use this solution with a natural pH for the synthesis of NPs without adding an acid or alkali?
3) After monitoring the synthesis of NPs, were they somehow purified or not, otherwise, the synthesis probably continued further? Then it is necessary to control the stability of the NPs (at least by PPR) before using the particles
4) In the section 3.4.1, it is strange that the antimicrobial activity of extract was not studied. There are some articles which is described the antimicrobial activity of Ginkgo biloba extracts (Sati SC, Joshi S. Antibacterial activities of Ginkgo biloba L. leaf extracts. Scientific World Journal. 2011;11:2241-6. doi: 10.1100/2011/545421. Cong Wang et al. Evaluation of the antimicrobial function of Ginkgo biloba exocarp extract against clinical bacteria and its effect on Staphylococcus haemolyticus by disrupting biofilms, Journal of Ethnopharmacology, 2022, V 298, 115602. https://doi.org/10.1016/j.jep.2022.115602.)
5) How NPs samples were prepared before use for microbiological research and how their concentration was determined?
6) Despite the fact that the extract does not exhibit antibacterial properties (by agar well diffusion method), the authors suggest that the resulting NPs are effective, including due to the activity of the biomaterial that stabilizes the NPs; in this regard, it would be interesting to analyze the activity of NPs with one concentration of silver but different concentrations of stabilizing agent.
Finally, despite the questions raised, I think that the article can be publish in the present form.
Author Response
Evaluation:The article of of Qi Ni et al described very interesting and voluminous research, in which the AgNPs synthesized and stabilized with the leaf extract of Ginkgo biloba were studied as antimicrobial agents. The work was performed with the use of wide range of research methods. The synthetic procedures, the characterization of obtained NPs, and antibacterial and cytotoxic properties of AgNPs is perfectly described. Thus, my questions to the authors have only debatable character and hand also touches on some experimental details.
A: Thank you very much for your review and for acknowledging our work. Your comments have been immensely helpful to our work. They have not only highlighted areas for improvement but also enhanced the quality of our manuscript. We will meticulously address and incorporate each of your suggestions in our revisions. Once again, we greatly appreciate your valuable feedback.
- In works related to the use of biological objects, the question always arises: will the composition of the extract change depending on the batch of the biological product and the place of origin, and how will this affect the final stage of use?
A1:Thank you for your question. This issue is indeed an unavoidable part. However, we can reduce errors caused by bioproducts by selecting the source of raw materials and minimizing errors during the experiment. As the main component for synthesizing AgNPs from Ginkgo biloba leaves is flavonoids. We found through literature review that the content of this substance is highest at the end of May throughout the year, so we collected and stored the materials at that time (Please see doi.org/10.3969/ j.issn.1000-2006.201803004). As for the source of the raw materials, we selected the nearest location and our analysis of plant metabolites showed a rich content of flavonoids, ensuring the credibility of the results. In future improvements, we can select areas with the highest content as the raw material collection site, such as Hanzhong, Shaanxi. In the experiment, we optimized the conditions to minimize differences caused by the raw materials. Strengthening the conditions can accelerate the reaction process, and the impact of multiple factors weakens the effect of a single factor. In conclusion, controlling the differences caused by the complex composition of natural products is indeed a challenge. Although differences are inevitable, and we can only strive to optimize and minimize them as much as possible.
- What is the pH of the extract solution and is it possible to use this solution with a natural pH for the synthesis of NPs without adding an acid or alkali?
A2:Thank you for your question. The natural pH of the plant extract solution is measured to be around 6.0, which is acidic. In our optimization of the conditions, we assessed the influence of pH on the synthesis of AgNPs. The results indicated that an alkaline environment is more conducive to the synthesis. At a pH of 6.0, the formation of AgNPs was not significant. Therefore, after screening, the optimal condition chosen was pH=9.
- After monitoring the synthesis of NPs, were they somehow purified or not, otherwise, the synthesis probably continued further? Then it is necessary to control the stability of the NPs (at least by PPR) before using the particles
A3:Thank you for your question. After the synthesis was complete, AgNPs wasn’t purified or other treatment. We just washed the product three times with deionized water at 11,000 rpm and then collected it by drying in a 60 °C vacuum oven. The dried product was weighed and dissolved for use in subsequent experiments.
- In the section 3.4.1, it is strange that the antimicrobial activity of extract was not studied. There are some articles which is described the antimicrobial activity of Ginkgo biloba extracts (Sati SC, Joshi S. Antibacterial activities of Ginkgo biloba L. leaf extracts. Scientific World Journal. 2011;11:2241-6. doi: 10.1100/2011/545421. Cong Wang et al. Evaluation of the antimicrobial function of Ginkgo biloba exocarp extract against clinical bacteria and its effect on Staphylococcus haemolyticus by disrupting biofilms, Journal of Ethnopharmacology, 2022, V 298, 115602. https://doi.org/10.1016/j.jep.2022.115602.)
A4:Thank you for your question and references. Considering the concepts of environmental protection and simplicity in green synthesis, we avoided complex extraction processes and the use of toxic chemical reagents. In our experiments, we only used aqueous extracts from Ginkgo biloba leaves. However, the content of active components in the aqueous extract might be lower than that in organic extracts. We found that in the referenced literature, antimicrobial experiments were carried out using a concentrated organic extract from a large quantity of Ginkgo biloba leaves, which had a much higher content of various phytochemicals. Moreover, in the article about antimicrobial activity of the outer skin of Ginkgo biloba fruits, they discovered that the supernatant of Ginkgo biloba fruit extract had no significant effect on certain clinical strains, including the E. coli and P.aeruginosa strains we used. Therefore, antimicrobial activity may be related to the strain type and concentration. In the manuscript, we use Ginkgo biloba leaves solely as reducing and capping agents to mediate the synthesis of AgNPs. During synthesis, we adopt a lower concentration of the leaf extract at 10 mg/ml. Thus, we focus the antimicrobial attention on the silver nanoparticles, where the low concentration of phytochemicals may play a supportive role in antimicrobial activity.
- How NPs samples were prepared before use for microbiological research and how their concentration was determined?
A5:Thank you for your question. In the microbiological experiments, the dried AgNPs particles collected after centrifugation were weighed and dissolved in sterile water for use. Different concentrations were prepared by serial dilution from the highest concentration to reduce errors.
- Despite the fact that the extract does not exhibit antibacterial properties (by agar well diffusion method), the authors suggest that the resulting NPs are effective, including due to the activity of the biomaterial that stabilizes the NPs; in this regard, it would be interesting to analyze the activity of NPs with one concentration of silver but different concentrations of stabilizing agent.
A6:Thank you for your suggestion. Since the antibacterial properties of Ginkgo biloba leaves have already been established in previous research, we did not conduct independent experiments. The lack of an inhibition zone in the extract might be due to the concentration's bacteriostatic effect being insufficient to inhibit microbial proliferation. Exploring different concentrations of stabilizers could be attempted. We are grateful for the ideas you have provided for our subsequent experiments.

Reviewer 2 Report
Comments and Suggestions for Authors
How much plant matter was collected?
The study of antimicrobial activity is an interesting way. How many repated were evaluated? It is not mentioned in the manuscript. Can you explain?
Why weren't classical methods evaluated for antimicrobial activity mentioned for example by EUCAST?
Why was negative control LB medium and bacteria used? It is illogical.
In all tests it is necessary to mention how many times the evaluation is repeated.
The manuscript is very well described but still has some flaws which must be corrected.
Author Response
Thank you for your review. Your comments have been immensely helpful to our work. They have enhanced the quality of our manuscript. We will meticulously address and incorporate each of your suggestions in our revisions. Once again, we greatly appreciate your valuable feedback.
- How much plant matter was collected?
A1:Thank you for your question. In the HPLC-MS analysis of Ginkgo biloba leaves, a total of 2552 compounds were detected. We selected only the top thirty identified compounds based on ion intensity for analysis. These compounds were sorted by intensity from highest to lowest to create Table 1.
- The study of antimicrobial activity is an interesting way. How many repated were evaluated? It is not mentioned in the manuscript. Can you explain?
A2:Thank you for your question. This was indeed an oversight. We only provided a general explanation in the data analysis section. Our antimicrobial activity experiments were carried out at least in triplicate. For maximum bactericidal concentration (MBC) assays, we plated three parallel samples for each concentration. The growth curves were executed in 96-well plates, with three wells per concentration. Cultures were grown in an automatic microplate reader at 37 °C, with the OD recorded every 15 min. Experiments were repeated three times with similar results. Data from three replicate sets were collected and analyzed using GraphPad for plotting. The inhibition zones of the antimicrobial circles were also determined by calculating the mean ± standard error of the mean from three replicate sets.
- Why weren't classical methods evaluated for antimicrobial activity mentioned for example by EUCAST?
A3:Thank you for your question. For the assessment of in vitro antimicrobial activity of plant extracts or purified compounds, the most common and basic methods are the disk diffusion and the broth or agar dilution methods. (Please see doi.org/10.1016/j.jpha.2015.11.005) We utilized the broth dilution and agar well diffusion methods. Agar well diffusion is one among several common diffusion methods, with a procedure similar to that of disk diffusion. Broth dilution is also one of the most fundamental methods for antimicrobial drug sensitivity testing. Moreover, to understand the dynamic interaction between antimicrobial agents and microbial strains, we examined the changes in antimicrobial activity over time. Although some techniques have been standardized by EUCAST, testing natural products often requires some modifications to the standardized protocols. Therefore, we did not strictly follow the EUCAST procedures but instead used methods commonly referenced in the literature. (Please see doi.org/10.2147/IJN.S251174, doi.org/10.1016/j.msec.2018. 03.035, doi.org/10.1016/j.envres.2023.115614)
- Why was negative control LB medium and bacteria used? It is illogical.
A4:Thank you for pointing out my mistake. The LB medium containing the bacterial fluid should serve as a positive growth control. We have made the following modifications in the manuscript: “LB broth containing the bacterial strains served as the positive control, while gentamicin and ampicillin were used as negative controls.” Please see Line 180-181. “Positive control wells contained 10 μL of liquid LB medium, while negative control wells contained antibiotics.” Please see Line 191-192.
- In all tests it is necessary to mention how many times the evaluation is repeated.
A5:Thank you for your suggestion. Each experiment was carried out at least in triplicate. We have added the description of experimental replicates in the manuscript. Please see Line 141-142, 186, 203, 230-231, 236-237.

Reviewer 3 Report
Comments and Suggestions for Authors
I would like to congratulate the authors for this nice piece of work. The manuscript is well written, quite designed and easy to read. The interpretations are quite good and supported by pertinent reference and the figures are well constructed. Minor comments could be considered as follows:
1. The abbreviations should spell our when mentioning for the first time.
2. What software did you use for statistical analysis?
3. The authors are encouraged to add a figure, illustrating the antibacterial mechanisms of the AgNPs against the studied bacteria.
4. The authors should illustrate the significant differences in Table 2 based on statistical analyses.
Author Response
Evaluation:I would like to congratulate the authors for this nice piece of work. The manuscript is well written, quite designed and easy to read. The interpretations are quite good and supported by pertinent reference and the figures are well constructed. Minor comments could be considered as follows.
A: Thank you for your interest in our manuscript content and the effort you put into reviewing it. We have thoroughly reviewed the valuable comments you provided and have made the necessary modifications as a result. Your constructive feedback has been immensely valuable to us.
- The abbreviations should spell our when mentioning for the first time.
A1:Thank you for your suggestion. Due to the word limit of Abstract, we delete the abbreviation of HPLC-MS and rewrite this sentence. Please see Line 16. We also check other abbreviations in our manuscript and correct them. Please see Line 41. 169, 194.
- What software did you use for statistical analysis?
A2:Thank you for your question. All the data were analyzed and plotted using Origin 2021 and GraphPad Prism 8.0.1. We have added the following description to lines 233-234 of the manuscript: “Data analysis and graphing were conducted using the Origin 2021 and GraphPad Prism 8.0.1 software.” Please see Line 233-234.
- The authors are encouraged to add a figure, illustrating the antibacterial mechanisms of the AgNPs against the studied bacteria.
A3:Thank you for your suggestion. We added a figure illustrating the mechanism of the antibacterial activity of AgNPs is displayed as Figure 8. Please see Line 454,455,477,478.
Figure 8. The probable mechanism of the antibacterial activity of AgNPs.
- The authors should illustrate the significant differences in Table 2 based on statistical analyses.
A4:Thank you for your suggestion. Due to the numerous changes in conditions, to make it clearer for readers to understand the variations within the conditions, we have created Table 2, which displays the specific locations of the absorption peaks. Labeling each absorption peak with its value on the figure would result in clutter. The selection of conditions was judged comprehensively based on the absorption values, peak positions, and shapes of the peaks. Table 2 can serve as a supplementary reference to Figure 3, primarily analyzing the final results presented by each condition in Figure 3.
